# Single-cell variability in multicellular organisms

Stephen Smith[1] & Ramon Grima[1]

Noisy gene expression is of fundamental importance to single cells, and is therefore widely studied in single-celled organisms. Extending these studies to multicellular organisms is challenging since their cells are generally not isolated, but individuals in a tissue. Cell–cell coupling via signalling, active transport or pure diffusion, ensures that tissue-bound cells are neither fully independent of each other, nor an entirely homogeneous population. In this article, we show that increasing the strength of coupling between cells can either increase or decrease the single-cell variability (and, therefore, the heterogeneity of the tissue), depending on the statistical properties of the underlying genetic network. We confirm these predictions using spatial stochastic simulations of simple genetic networks, and experimental data from animal and plant tissues. The results suggest that cell–cell coupling may be one of several noise-control strategies employed by multicellular organisms, and highlight the need for a deeper understanding of multicellular behaviour.

---

[1] School of Biological Sciences, University of Edinburgh, Mayfield Road, Edinburgh, EH9 3JR Scotland, UK. Correspondence and requests for materials should be addressed to R.G. (email: ramon.grima@ed.ac.uk)

I t is now well established that stochastic gene expression is the main driver of phenotypic variation in populations of genetically identical cells[1,2]. In populations of single-celled organisms, individuals are known to switch between metabolic states[3] or antibiotic resistant states[4], and to randomly choose the timing of reproduction[5], among other stochastic survival strategies. The availability of single-cell fluorescence data has precipitated a wealth of mathematical modelling approaches to understand single-cell noise based on the chemical master equation (CME)[6], such as the stochastic simulation algorithm (SSA)[7], the finite-state projection algorithm (FSP)[8], and the linear noise approximation (LNA)[9,10].

In multicellular organisms, mouse olfactory development[11] and *Drosophila* vision[12] are well-known examples of stochastic gene expression in tissues, along with pattern formation[13,14] and phenotypic switching of cancer cells[15]. More recently, it has been observed that tissue-bound cells can take advantage of polyploidy to reduce noise[16]. Nevertheless, single-cell variability in tissues is considerably less well understood than in isolated cells, for two main reasons.

Firstly, acquiring fluorescence data for tissue-bound cells requires a combination of high-resolution imaging and cell segmentation software that has only recently become possible for mRNA localisation[17] and still poses a significant challenge for proteins. The difficulty of accurate segmentation of tissue-bound cells means that the majority of segmented time course data still concerns populations of isolated cells[18], while tissue-level data has historically been too low-resolution to distinguish individual cell outlines[19], though improvements in microscopy are increasingly eliminating this problem[16].

Secondly, the transfer of material between tissue-bound cells makes mathematical modelling of tissues significantly more complex than equivalent isolated cell models. In addition to the long-range endocrine networks which connect all cells in a tissue, neighbouring cells communicate via complex paracrine signalling networks[20], and also via small watertight passages such as gap junctions in animals, and plasmodesmata in plants. In plant cells, molecules up to and including proteins are known to move through plasmodesmata by pure diffusion[21,22], while those as large as mRNA are actively transported[23]. In animal cells, peptides diffuse through gap junctions[24], while larger molecules have been shown to be transported across cytoplasmic bridges[25] or tunnelling nanotubes[26]. A single cell in a tissue is therefore partially dependent on its neighbour cells, but also partially independent of them, and so mathematical models of cells within multicellular organisms must take account of this coupling.

In this article, we start from a general mathematical description of a tissue of cells, in which each cell contains an identical stochastic genetic network, with identical reaction rates. Our description permits molecules to move from a cell to a neighbouring cell with a given transport rate or coupling strength, representing signalling, active transport, or pure diffusion. We subsequently consider two special cases: when the coupling is very weak and very strong. In both of these cases, our complex mathematical description reduces to simple expressions for the single-cell variability. These equations are completely generic, and apply to any biochemical network including oscillatory and multimodal systems.

The implication of the equations is that single-cell variability is controlled by the strength of cell–cell coupling, in a manner that depends on the Fano factor (FF) of the underlying genetic network. If $FF > 1$, then cell–cell coupling will tend to reduce the single-cell variability (or equivalently, the heterogeneity of the tissue); whereas if $FF < 1$, then coupling will tend to increase the single-cell variability. To confirm our theory, we use spatial stochastic simulations of three biochemical networks, and

experimental data from rat pituitary tissue, a leaf of *Arabidopsis thaliana*, and a population of mouse fibroblast cells.

## Results

**Illustrative examples**. Modelling approaches to genetic networks such as the CME, SSA, or LNA assume that the cell is an isolated volume with no molecules entering or leaving the system from outside (Fig. 1a). Tissues of cells violate this assumption: each cell is connected to a number of neighbour cells (Fig. 1b), and molecules involved in the genetic network can be transported from cell to cell. The differences between a population of identical independent cells and a tissue of identical connected cells can be seen with stochastic simulations of a simple genetic network. In Fig. 1c we plot three independent realisations of the SSA for the well-known two-stage gene expression network[6]:

$$\emptyset \underset{d_0}{\overset{v_0}{\rightleftharpoons}} M, M \xrightarrow{v_1} M + P, P \xrightarrow{d_1} \emptyset, \qquad (1)$$

in which a molecule of mRNA ($M$) is transcribed with rate $v_0$ and decays with rate $d_0$. The mRNA can translate a protein ($P$) with rate $v_1$ which in turn decays with rate $d_1$. The trajectories in Fig. 1c correspond to the number of protein molecules in three independent cells.

To model a tissue, we imagine a $N \times N$ grid of cells (Fig. 1b) numbered from 1 to $N^2$ with the genetic network (1) inside each cell. In addition, we couple neighbouring pairs of cells by allowing the protein $P$ to be transported between them with a rate $t$. To model this, we think of protein transport from cell $i$ to cell $j$ as a simultaneous decay of protein in cell $i$ and creation of protein in cell $j$. Specifically, we can write the system in cell $i$ as:

$$\emptyset \underset{d_0}{\overset{v_0}{\rightleftharpoons}} M_i, M_i \xrightarrow{v_1} M_i + P_i, P_i \xrightarrow{d_1} \emptyset, P_i \underset{t}{\overset{t}{\rightleftharpoons}} P_j, \qquad (2)$$

where $M_i$ and $P_i$ denote the mRNA and protein respectively in cell $i$, and the reaction $P_i \overset{t}{\underset{t}{\rightleftharpoons}} P$ denotes the transport of protein from cell $i$ to cell $j$, if $i$ and $j$ are neighbouring cells. Transport is therefore modelled as a kind of 'reaction' involving two species $P_i$ and $P_j$, though biologically these are really the same species in different locations. We note that this model of transport implies exponentially distributed waiting times between successive transport events, an assumption that has previously been used when modelling active transport in tissues[27,28], and in modelling reaction-diffusion systems[29,30]. The main results of this article do not depend on the exponential assumption since we only analyse the fast limit of transport, though it is convenient for simulations at finite transport rates.

This description of transport has a clear advantage for modelling: we have reframed a complex problem of cell–cell transport into a simpler problem of species and reactions on which, in principle, we can use the FSP, SSA, or even LNA. In reality, the FSP and LNA are impractical for such systems, owing to their large dimensions: for a tissue with 100 cells, the system (2) consists of many more species than system (1) (200 rather than 2), many more chemical reactions (400 rather than 4) and many additional transport "reactions" (roughly 400, rather than 0). The SSA, however, is still a useful technique for getting accurate data about tissue systems like 2, though it will obviously be substantially slower than for single-celled systems like (1).

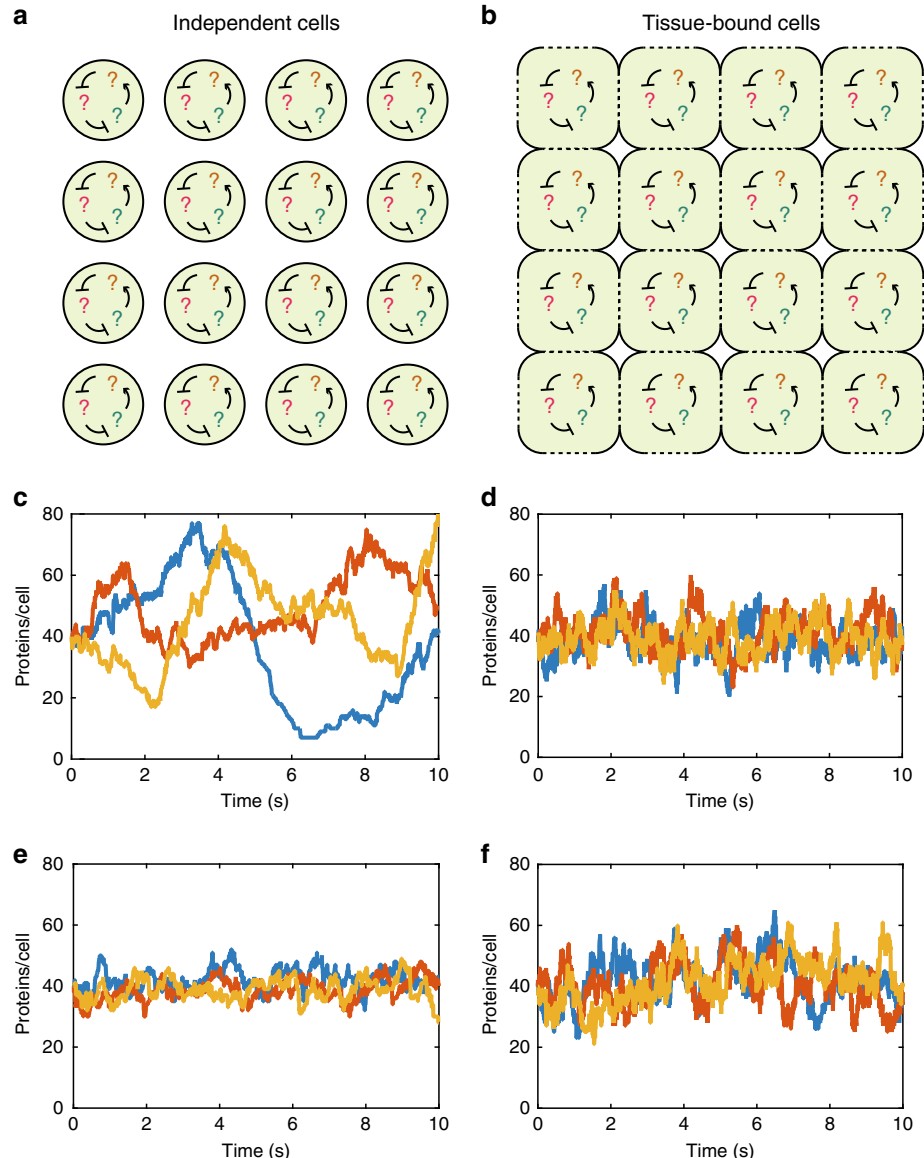

**Fig. 1** Differences between a population of isolated cells and a tissue of cells. **a** A population of isolated cells: each cell contains an identical genetic network. **b** A tissue of cells: each cell contains an identical genetic network and some molecules can be transported between neighbouring cells (dotted lines). **c** Typical single-cell protein trajectories of system (1) in isolated cells. **d** Typical single-cell protein trajectories of system (1) in a tissue of connected cells: noise is clearly reduced compared to **c**. **e** Typical single-cell protein trajectories of system (3) in isolated cells. **f** Typical single-cell protein trajectories of system (3) in a tissue of connected cells: noise is clearly increased compared to **e**. Parameter values are $v_0 = 4$, $d_0 = 1$, $v_1 = 10$, $d_1 = 1$, $t = 10$, $N^2 = 100$, $V_C = 1$ for system (1) and $k_1 = 32$, $k_2 = 0.01$, $t = 10$, $N^2 = 100$, $V_C = 1$ for system (3)

We simulated system (2) with a version of the SSA[31] with $N^2 = 100$ cells, giving 100 trajectories of protein number, one for each cell. Three typical trajectories are plotted in Fig. 1d. Notably, the tissue trajectories in Fig. 1d are considerably less variable (more homogeneous) than the isolated cell trajectories in Fig. 1c.

This apparent increase in homogeneity is perhaps unsurprising, and may be thought of as the obvious consequence of increasing coupling. However, remarkably, coupling can also reduce the homogeneity in a tissue. For example, a simple system representing the synthesis of a protein $P$, and its consequent dimerisation into a homodimer $D$, is defined by the reactions:

$$\emptyset \xrightarrow{k_1} P, P + P \xrightarrow{k_2} D. \qquad (3)$$

In a tissue, this system involves the transport of $P$ between neighbouring cells. We simulated system (3) both without and

with transport using the same version of the SSA, and three typical single-cell trajectories of the protein $P$ are plotted in Fig. 1e, f respectively. The independent cell trajectories are relatively homogeneous, while the tissue-bound cell trajectories are substantially more variable (more heterogeneous).

An intuitive explanation can be made for these initially surprising observations. The transport of molecules between cells has two distinct effects on the single-cell variability: (1) by moving molecules into and out of cells, it allows for greater cell–cell variation; (2) by smoothing out concentration gradients between neighbouring cells, it homogenises concentrations across the tissue.

The effect of transport on single-cell variability is determined by the trade-off between effects (1) and (2). In Fig. 1c, the cells have large fluctuations in molecule number which will be reduced by effect (2), but new fluctuations will be induced by effect (1). In

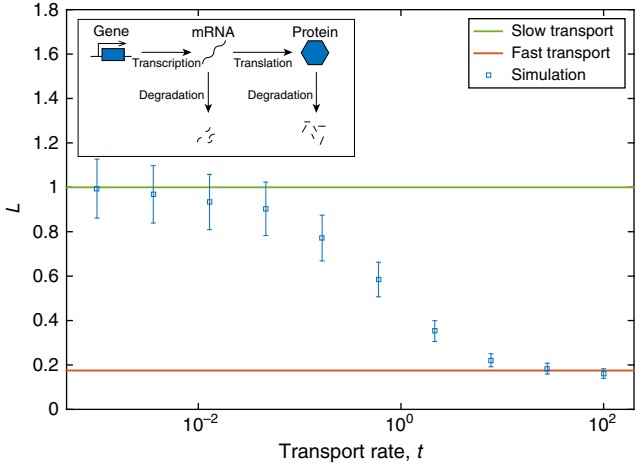

**Fig. 2** $L$ as a function of protein transport rate for the two-stage gene expression system (1). Theoretical values for the fast transport limit (red), and slow transport limit (green) are shown as solid lines. Simulation data is shown for the average single-cell variability (blue squares) for a variety of protein transport rates. Parameter values are $v_0 = 3$, $d_0 = 1$, $v_1 = 10$; $d_1 = 1$, $V_C = 1$, $V_T = 100$. Inset: schematic diagram of system (1)

Fig. 1d new fluctuations have been added, but these are not large enough to offset the reduction in the original fluctuations, and so, overall, homogeneity is increased by transport. Meanwhile, for system (3), effect (2) is much less significant because the single-cell variability in Fig. 1e is already small, so effect (1) dominates and there is an overall increase in heterogeneity in Fig. 1f.

**Theory**. To make this intuition mathematically precise (see Methods), we consider a system of any number of species and any number of reactions, a tissue with volume $V_T$, and a cell of volume $V_C$. Furthermore, we let $n$ be the number of molecules of a species of interest in the tissue, while $m$ is the number of molecules of that species in the cell. Note that because each cell contains the same genetic network, the variance of fluctuations in a single cell, $\langle m^2 \rangle - \langle m \rangle^2$, is a non-normalised measure of the single-cell variability. We define our measure of single-cell variability, $L$, to be:

$$L = \frac{V_T \left( \langle m^2 \rangle - \langle m \rangle^2 \right)}{V_C \left( \langle n^2 \rangle - \langle n \rangle^2 \right)}, \qquad (4)$$

i.e., the ratio of the variance of fluctuations in a single cell to the variance of fluctuations in the entire tissue, scaled by volume. The reason for this definition becomes clear when we consider the limiting case of weak cell–cell coupling (i.e., isolated cells). In this case, each cell is completely independent and hence it directly follows by the Bienaymé formula[32] (Methods section) that:

$$\left( \langle m^2 \rangle - \langle m \rangle^2 \right) = \frac{V_C}{V_T} \left( \langle n^2 \rangle - \langle n \rangle^2 \right), \qquad (5)$$

which immediately implies that $L = 1$. That is, if $L = 1$, then the cell-to-cell variability is at the level we would expect if the cells were completely independent of each other. This can be considered as a neutral state, neither particularly heterogeneous, nor especially homogeneous.

If $L < 1$, then the single-cell variability is lower than we would expect from the Bienaymé formula, given the tissue-level variance. It follows that the cells are more homogeneous than decoupled cells. On the other hand, if $L > 1$, then the cell-to-cell variability is higher than we would expect from the Bienaymé formula, given the tissue-level variance. It follows that the cells

are more heterogeneous than decoupled cells. $L$ is therefore a non-dimensional statistical measure of single-cell variability (or equivalently, population heterogeneity).

With this in mind, we next consider what happens to a tissue of cells with cell–cell coupling. At zero coupling, we will naturally have $L = 1$. As the coupling strength increases, $L$ will change, but the magnitude of the change will depend on a number of system-specific factors including the topology of the tissue (which cells are coupled to which), the structure of the genetic network, and the rates of the reactions involved. To bypass these issues, we consider the special case of infinitely fast cell–cell transport, and we reason that the behaviour at finite transport rates will lie between the zero coupling and infinite coupling cases.

Biologically we can think of infinite coupling as the extreme case where a protein will move from cell to cell many times during its lifetime. In this case, the probability of finding a given molecule in a given cell is simply $\frac{V_C}{V_T}$. This implies that the probability distribution governing the number of molecules in the cell is a convolution of the solution of the CME (which describes the whole tissue) and a binomial distribution (Methods section). While the convolution is generally impossible to solve, remarkably we can obtain a simple expression linking the variance of fluctuations in the cell, with the variance in the entire tissue:

$$\langle m^2 \rangle - \langle m \rangle^2 = \frac{V_C}{V_T} \langle n \rangle + \frac{V_C^2}{V_T^2} \left( \langle n^2 \rangle - \langle n \rangle - \langle n \rangle^2 \right). \qquad (6)$$

Combining Eqs. (4) and (6), and defining the Fano factor (FF) as the ratio of tissue-level variance to the mean, $FF = \frac{\langle n^2 \rangle - \langle n \rangle^2}{\langle n \rangle}$, we find that the single-cell variability at infinite coupling is given by:

$$L = \frac{V_C}{V_T} + \frac{1 - \frac{V_C}{V_T}}{FF}. \qquad (7)$$

We note now that FF is a standard statistical measure of the size of fluctuations. Probability distributions with $FF = 1$ are said to have Poissonian fluctuations, while $FF < 1$ corresponds to subpoissonian and $FF > 1$ to superpoissonian. Our earlier intuition suggested that systems with large fluctuations would tend to see a reduction in cell-to-cell variability as coupling strength increases. Now we see that this is indeed the case: combining $FF > 1$ with Eq. (7), we find that $L < 1$ at infinite coupling strength, suggesting that coupling tends to decrease single-cell variability for superpoissonian systems. Alternatively, choosing $FF < 1$ we find that $L > 1$ at infinite coupling strength, implying that coupling will increase single-cell variability for subpoissonian systems.

For system (1), the Fano factor can be computed exactly since the moment equations for the corresponding CME are closed[33]. In particular, we have that $FF = 1 + \frac{v_1}{d_0 + d_1} > 1$, implying that increasing the transport rate will reduce the single-cell variability, as shown in Fig. 1c, d.

For system (3), the presence of a bimolecular reaction prevents the moment equations from closing, and so the moments are instead obtained from the steady-state distribution of molecule numbers. The mean and variance are given in ref.[34], but we will not state them here since they are complicated expressions. Instead we note that $FF < 1$ for all parameter values, suggesting that the single-cell variability will increase as cell–cell transport increases. See the next section for more details of these calculations.

For these examples the qualitative changes in single-cell variability are independent of parameter values, though this would not be the case for systems with Fano factors which can vary from subpoissonian to superpoissonian. We note that these results are independent of the spatial structure of the tissue: they

apply equally to neighbour-neighbour and long-range interactions, and indeed any kind of coupling provided no cells in the tissue are disconnected from the population. We also stress that these results apply equally to systems out of equilibrium, including oscillatory systems and systems far from steady-state, since no assumptions have been made on the type of biochemical network inside each cell.

**Verification of theory using stochastic simulations.** Our theory pedicts that the single-cell variability $L$ should move from 1 to the value given in Eq. (7) as cell–cell transport increases. In this section we test the accuracy of this prediction on data from detailed stochastic simulations using a version of the SSA[31] that is well-suited to simulating tissues.

First, we again consider the two-stage gene expression system (1) as shown in Fig. 1c, d and Fig. 2 inset. Since the moments of the CME are closed for this system[33] we can find exact expressions for the tissue-level mean, $\frac{V_T v_0 v_1}{d_0 d_1}$, and the tissue-level variance, $\frac{V_T v_0 v_1}{d_0 d_1}\left(1 + \frac{v_1}{d_0 + d_1}\right)$. It follows that $FF = 1 + \frac{v_1}{d_0 + d_1}$, and so Eq. (7) implies that $L$ will decrease from 1 to $\frac{d_0 + d_1 + v_1 \frac{V_C}{V_T}}{d_0 + d_1 + v_1}$ as transport increases.

As a second example we consider the protein synthesis and dimerisation system (3) as shown in Fig. 1e, f and 3 inset. The mean and variance are given in ref.[34], and they imply that $FF = \frac{3}{4} - \phi \frac{I_1'(4\phi)}{I_1(4\phi)} + \frac{\phi}{\left(\frac{1}{4\phi} + \frac{I_1'(4\phi)}{I_1(4\phi)}\right)}$, where $\phi = V_T\sqrt{\frac{k_1}{2k_2}}$ and $I_1(x)$ is the modified Bessel function of the first kind[35] and $I_1'(x)$ is its derivative. Numerical analysis confirms that $FF < 1$ for all values of $\phi$, suggesting that $L$ will increase from 1 as cell–cell transport increases.

For our third example we consider the bimodal three-stage gene expression network studied in refs. 6,9 (Fig. 4 inset):

$$G_{on} \underset{k_{on}}{\overset{k_{off}}{\rightleftharpoons}} G_{off}, \quad G_{on} \xrightarrow{v_0^{on}} G_{on} + M, \quad G_{off} \xrightarrow{v_0^{off}} G_{off} + M,$$

$$M \xrightarrow{d_0} \emptyset, \quad M \xrightarrow{v_1} M + P, \quad P \xrightarrow{d_1} \emptyset, \tag{8}$$

in which a gene can be in an active state ($G_{on}$) or an inactive state ($G_{off}$). The active gene transcribes mRNA ($M$) with a rate $v_0^{on}$, while the inactive gene transcribes mRNA with a rate $v_0^{off}$. The protein is translated as in the earlier system (1). We again calculate the mean and variance of fluctuations for the protein $P$ from the moment equations, as for the previous examples, and we find that the Fano factor is larger than 1 so Eq. (7) again implies that $L$ will decrease as cell transport increases.

In summary, in Figs. 2, 3 and 4 we compare the analytical expressions for fast transport, Eq. (7), with the simulation data for systems (1), (3) and (8), respectively. It is clear that in every case our theoretical predictions are correct. For each example the single-cell variability, $L$, from simulations (blue squares) moves from the slow limit, 1, (green line) to the predicted fast transport limit (red line). As predicted by the Fano factor criterion, for systems (1) and (8) $L$ decreases with transport rate, while $L$ increases for system (3).

**Application to experimental data.** Testing our predictions on simulations is useful, because by varying the rate of transport we can confirm that increasing it leads to the predicted change in single-cell variability, but with experimental data the transport rate is both fixed and completely unknown. However, we know that $L$ lies between 1 and the value given by Eq. (7), and that the parameters of Eq. (7) are either tissue-level quantities (FF), or easily calculable ($V_T$ and $V_C$). It follows that we can use tissue-level time course data to estimate $L$, without any knowledge of the underlying genetic network. In general, the corresponding single-cell data would not be available, however we specifically choose examples with both tissue-level and single-cell data so as to check that our estimates are correct.

We first apply our method to fluorescence data of GFP concentration in two distinct rat pituitary tissues[20] in which cells communicate both via paracrine signalling and active transport across gap junctions. The fluorescence data is available at the single-cell level, so the tissue-level data is obtained simply by summing up the single-cell fluorescence. We apply our method to this tissue-level data, and subsequently check its accuracy using single cells.

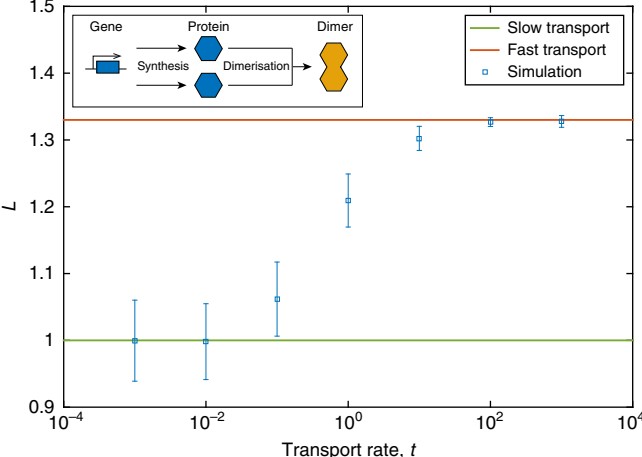

**Fig. 3** $L$ as a function of protein transport rate for the dimerisation system (3). Theoretical values for the fast transport limit (red) and slow transport limit (green) are shown as solid lines. Simulation data is shown for the average single-cell variability $L$ (blue squares) for a variety of protein transport rates. Parameter values are $k_1 = 32$, $k_2 = 0.01$, $V_C = 1$, $V_T = 100$. Inset: schematic diagram of system (3)

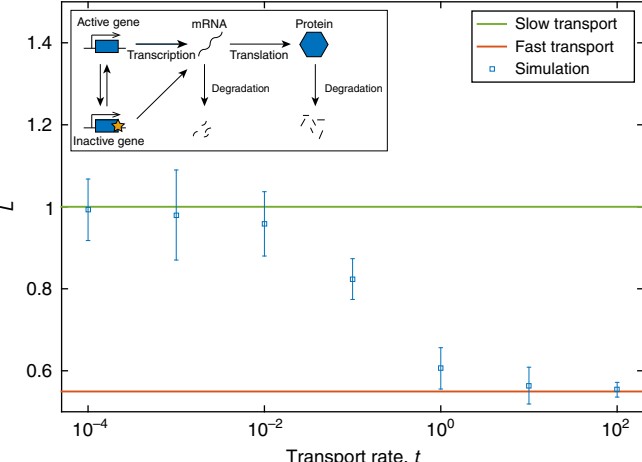

**Fig. 4** $L$ as a function of protein transport rate for the three-stage gene expression system (8). Theoretical values for the fast transport limit (red), and slow transport limit (green) are shown as solid lines. Simulation data is shown for the average single-cell variability (blue squares) for a variety of protein transport rates. Parameter values are $k_{on} = 0.1$, $k_{off} = 0.1$, $v_0^{on} = 3$, $v_0^{off} = 1$, $d_0 = 1$, $v_1 = 1$, $d_1 = 1$, $V_C = 1$, $V_T = 25$. Inset: schematic diagram of system (8)

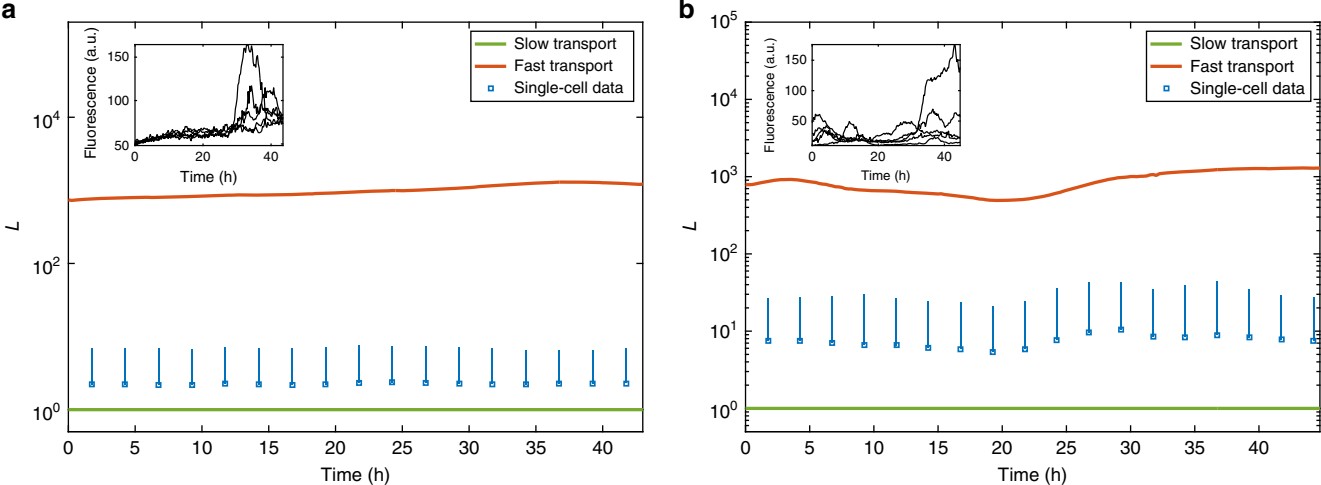

**Fig. 5** Comparison of fast (red) and slow (green) transport limits with single-cell data (blue squares: mean; blue bars: 1 standard deviation above mean) for **a** a tissue of 117 E18.5 rat pituitary cells, **b** a tissue of 114 P1.5 rat pituitary cells. Insets: typical single-cell trajectories from the raw data. Data are taken from ref. [20]

The first tissue is taken from a day 18.5 embryonic rat (E18.5), where cell–cell junctions are rare and their related proteins (E-, N-cadherin and β-catenin) and paracrine signalling proteins are expressed at a low level; while the second tissue is taken from a day 1.5 post-natal rat (P1.5) where junctions are considerably more common, and there is a high level of expression of related proteins[20]. The authors of ref. [20] note that the P1.5 tissue, while clearly more mature than the E18.5 tissue, has not yet reached the level of connectivity of the adult tissue for which the number of gap junctions is likely to be even higher. With this in mind, we expect $L$ in the E18.5 cells to be noticeably closer to 1 than the P1.5 cells, but the P1.5 cells should not be too close to (7) since they still are not fully mature.

In Fig. 5a, b we plot the fast and slow transport limits (green and red lines), and $L$ averaged over each cell (blue squares) and bars representing one standard deviation above the mean (blue bars) for the E18.5 and P1.5 tissues respectively. As expected, $L$ remains between the two in both cases, but is noticeably closer to 1 in Fig. 5a than in Fig. 5b.

The above dataset is further confirmation of our theory, but both it and the simulated systems are either in equilibrium or approaching it. Since this is frequently not the case in reality, we now apply our method to two oscillating datasets, one which we expect to have fast transport and one with slow transport. We stress that our method should apply to oscillatory systems since we have made no assumptions about the underlying genetic networks—only that the same genetic networks should be present in each cell, with the same reaction rates.

The second dataset corresponds to luminescence data of an oscillating protein concentration in a single leaf of *Arabidopsis thaliana*[19]. The luminescence data is available in image form, in which each pixel is close to single-cell resolution, so we can apply our method to the whole-leaf protein trajectory and subsequently check its accuracy with the single-pixel data.

In Fig. 6a we plot the slow and fast transport limits (green and red lines respectively) over time, and also $L$ averaged over each pixel of the leaf image (blue squares) and bars representing one standard deviation above the mean (blue bars). Since proteins are frequently transferred between cells in a plant tissue, we might expect $L$ to remain between the two limits but close to the fast limit, and such proves to be the case.

The third dataset consists of luminescence data of an oscillating protein concentration in a small population of mouse fibroblast cells[36]. The cells were imaged on the same plate for a period of over a month, and were sufficiently far apart that single-cell resolution is easily possible. The cells therefore do not exactly form a tissue (though we might expect some very low-level exchange of material) so we expect $L$ to be close to 1.

In Fig. 6b we plot the limits (green and red lines), and $L$ averaged over each cell (blue squares) and bars representing one standard deviation above the mean (blue bars). As expected, $L$ remains between the two, but is significantly closer to 1 than to the fast limit.

## Discussion

Single-cell variability in tissues is an immediate consequence of the stochastic nature of gene expression, but it can have significant phenotypic implications ranging from pattern formation to cancer. The cell–cell coupling characteristic of tissues ensures that the question of single-cell variability will be more complex than in the well-studied case of noise in single-celled organisms. As an attempt to address this question, we introduced a measure of single-cell variability, $L$, which is a non-dimensional statistical coefficient which determines whether the cells in a tissue are more or less heterogeneous than an equivalent population of independent cells. We found that $L$ is sensitive to the strength of coupling between cells in the tissues, in a manner that depends on the statistics of the underlying genetic network.

In the case of biochemical systems which naturally have large stochastic fluctuations (superpoissonian systems), we showed that increasing the coupling strength will tend to decrease the single-cell variability. On the other hand, for systems with small stochastic fluctuations (subpoissonian systems), we found that increasing the coupling strength will tend to increase the single-cell variability. These predictions were confirmed with stochastic simulations of simple genetic networks, and experimental data from both animal and plant tissues. These results suggest that cell–cell coupling could be one of several techniques cells use to control noise, while also highlighting the need for much greater understanding of multicellular behaviour.

We note here that we have focussed on fluctuations caused by the stochastic nature of biochemical reactions (intrinsic noise) and not noise induced by fluctuations in the environmental conditions (e.g., light level, temperature). This is because environmental fluctuations affect each cell in the population equally[37], and so will not affect the heterogeneity of the population.

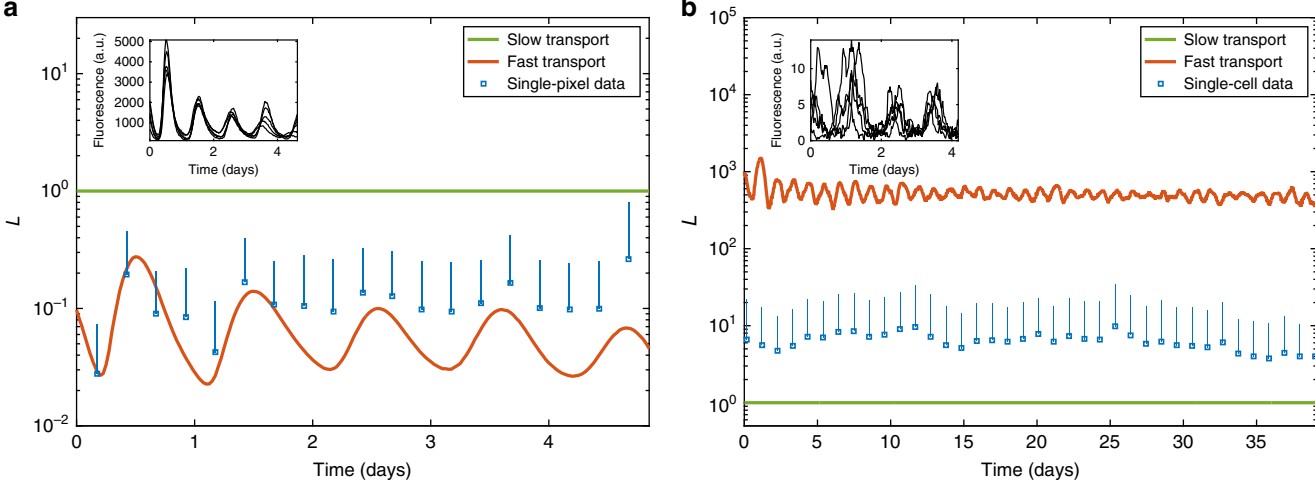

**Fig. 6** Comparison of fast (red) and slow (green) transport limits with single-pixel data (blue squares: mean; blue bars: 1 standard deviation above mean) for **a** a single leaf of *Arabidopsis thaliana*, **b** a population of 30 mouse fibroblast cells. Insets: typical single-cell trajectories from the raw data. Data is taken from **a** ref. [19], available at ref. [39], **b** ref. [36]

We also note that, although we are interested in heterogeneity, we have concentrated here on homogeneous tissues, that is, tissues where each cell contains an identical genetic network. We have found that it may be possible to extend our results to tissues with heterogeneous populations of cells, where different cells could have different noise statistics, though the relevant analyses are more complex than those in this article. This extension of our results would have fascinating applications to tissue ageing[38] and tumour growth, and will be the subject of a future paper.

## Methods

**Derivation of fast and slow limits of L**. We consider a tissue of $N^2$ cells with volume $V_T$, and a single cell with volume $V_C = V_T/N^2$, as well as identical systems of $M$ chemical species $X_1, \ldots, X_M$ interacting in each cell. Let $P(\vec{n}; V_T)$ be the probability that there are $\vec{n} = (n_1, \ldots, n_M)$ molecules of $X_1, \ldots, X_M$ respectively in the entire tissue. The tissue-level Fano factor of species $X_j$ is defined as $\mathrm{FF}_j = \frac{\langle n_j^2 \rangle - \langle n_j \rangle^2}{\langle n_j \rangle}$. Now, let $\vec{m} = (m_1, \ldots, m_M)$ be the number of molecules of $X_1, \ldots, X_M$, respectively, in the single cell, and let the corresponding probability distribution be $Q(\vec{m}; V_C)$.

If transport is slow, the system in each cell is independent of the rest of the population, and so, by the Bienaymé formula[32], the sum of the variances in each cell is equal to the variance in the tissue. The variance in a single cell is then given by, $\langle m_j^2 \rangle - \langle m_j \rangle^2 = \left[ \langle n_j^2 \rangle - \langle n_j \rangle^2 \right]/N^2 = (V_C/V_T)\left[ \langle n_j^2 \rangle - \langle n_j \rangle^2 \right]$. Furthermore, since all cells are statistically identical the mean concentration in each cell is the same and equal to that of tissue, $\langle m_j \rangle / V_C = \langle n_j \rangle / V_T$. It follows immediately from these considerations that $L = 1$.

For the fast transport limit we can relate the local solution $Q$ to the global solution $P$ using the theorem of total probability, $Q(\vec{m}; V_C) = \sum_{\vec{n}=0}^{\infty} Pr(\vec{m}|\vec{n}; V_C, V_T)P(\vec{n}; V_T)$, where the notation $\sum_{\vec{n}=0}^{\infty}(\cdot)$ is shorthand for $\sum_{m_1=0}^{\infty} \cdots \sum_{m_M=0}^{\infty}$, and $Pr(\vec{m}|\vec{n}; V_C, V_T)$ is the probability of finding $\vec{m}$ molecules of $X_1, \ldots, X_M$ respectively in $V_C$ given that there are $\vec{n}$ molecules respectively in $V_T$.

The limit of fast transport implies that molecules move into and out of the the cell much more frequently than they are involved in reactions. The molecules are uniformly distributed in $V_T$ under these conditions, so that the probability that a randomly chosen molecule is in $V_C$ is simply $\frac{V_C}{V_T}$. It follows from combinatorics that the probability of finding $m_j$ molecules of species $X_j$ in $V_C$ given that there are $n_j$ in $V_T$ is $(n_j!/(m_j!(n_j - m_j)!)) (V_C/V_T)^{m_j}(1 - V_C/V_T)^{n_j - m_j}$, that is, a Binomial distribution. It follows that $Pr(\vec{m}|\vec{n}; V_C, V_T)$ is the product of the mass functions of $M$ Binomial $(n_j, V_C/V_P)$ distributions for each species $X_j$. An expression for the single-cell distribution $Q$ in terms of the global distribution $P$ can then be written,

$$Q(\vec{m}; V_C) = \sum_{\vec{n}=0}^{\infty} P(\vec{n}; V_T) \prod_{j=1}^{M} \left[ \binom{n_j}{m_j} \left(\frac{V_C}{V_T}\right)^{m_j} \left(1 - \frac{V_C}{V_T}\right)^{n_j - m_j} \mathbf{1}_{\vec{m} \leq \vec{n}} \right]$$

The indicator function $\mathbf{1}_{\vec{m} \leq \vec{n}}$ prevents the expression from evaluating the impossible probabilities of finding more molecules in $V_C$ than in $V_T$, and therefore permits us to sum from zero to infinity without worry. Since this equation gives the single-cell distribution $Q$, we can use it to evaluate the single-cell second moment

which is given by $\langle m_j^2 \rangle = \sum_{\vec{m}=0}^{\infty} m_j^2 Q(\vec{m}, V_C)$. Swapping the summations over $\vec{n}$ and $\vec{m}$, and absorbing the indicator function into the latter summations, gives:

$$\langle m_j^2 \rangle = \sum_{\vec{n}=0}^{\infty} P(\vec{n}; V_T) \times$$
$$\left[ \sum_{m_1=0}^{n_1} \cdots \sum_{m_M=0}^{n_M} m_j^2 \prod_{k=1}^{M} \binom{n_k}{m_k} \left(\frac{V_C}{V_T}\right)^{m_k} \left(1 - \frac{V_C}{V_T}\right)^{n_k - m_k} \right]. \quad (9)$$

The local second moment is therefore the expected value of the quantity in square brackets under the global distribution $P$. The quantity in square brackets, however, is merely the expected value of $m_j^2$ under the $M$ independent Binomial$(n_j, V_C/V_T)$ distributions, and is therefore equal to $n_j \left(\frac{V_C}{V_T}\right)\left(1 - \frac{V_C}{V_T}\right) + n_j^2 \frac{V_C^2}{V_T^2}$. It follows that the local second moment is simply given by $\langle m_j^2 \rangle = \frac{V_C}{V_T} \langle n_j \rangle + \frac{V_C^2}{V_T^2} \langle n_j^2 \rangle - \frac{V_C^2}{V_T^2} \langle n_j \rangle$ It subsequently follows that the fast transport limit of $L$ has the form of Eq. (7).

**Data analysis**. Fluorescence trajectories are passed through a moving average filter with a window size dependent on the time-resolution of the data. The smoothed trajectory is considered to be a time-dependent mean, $\langle n \rangle$. Subtracting the mean from the raw trajectory gives a stationary noise component. The variance of this component is used to obtain a time-dependent estimate for $L$ or FF.

**Data availability**. All relevant data are available from the authors.

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

## Acknowledgements

This work was supported by a BBSRC EASTBIO PhD studentship to S.S. and by a Leverhulme grant award RPG-2013-171 to R.G. We thank Andrew Millar and Uriel Urquiza for useful discussions.

## Author contributions

S.S. designed research, carried out research and wrote the manuscript. R.G. designed research and helped write the manuscript.

## Additional information

**Competing interests:** The authors declare no competing financial interests.

