## [Peer Review File · Nature Communications]

Reviewers' comments:

Reviewer #1 (Remarks to the Author):

The paper studies expression noise in multicellular organisms where cells are grouped in tissues. Within a tissue, transport of molecules between individual cells takes place as opposed to single-cell analysis wherein each cell is independent of others. Overall, the paper studies an interesting topic, but the math analysis seems too simplistic, and noise insights known.

1) Is it a fair assumption that a complex process like transportation is exponentially distributed? Are the results dependent on this assumption? Will it be more realistic to assume transportation between neighboring cells, rather than any two cells? Given the spatial structure of the tissue, a specific cell may only be able to communicate with a few cells in its vicinity. Unless this is taken into account, it appears that their formulation may not be able to quantify the noise in individual cells.

2) An exact similar results on increase and decrease in noise depending on Fano factors has also been shown in other context, see Burger et al, PRE 2012. Influence of decoys on the noise and dynamics of gene expression

3) Although authors comment on noise in slow/fast transport, they do not comment upon the behavior at intermediate transport rates. Moreover, it will be useful to comment how different noise mechanisms (external disturbances vs. low copy effects) may change the results.

4) I was surprised to see no mention of paracrine signaling, which is the canonical cell-to-cell communication. In this case secreted signals, activate expression in other cells that lead to more secretion, creating spatial feedbacks. It will be useful to comment on such cases, which are far more common than those mentioned

5) The data analysis was confusing. My understanding is that the simple expression models don't show oscillations, and system being analyzed does oscillate. It is not clear from the analysis how the noise results are true for any architecture with any form of dynamics.

Reviewer #2 (Remarks to the Author):

In this paper Grima and colleagues analyze the impact of inter-cellular transport of proteins in tissues on the single cell noise levels. Using simple intuitive arguments and rigorous mathematics and simulations they demonstrate that exchange of material in tissues can reduce single cell variability for super-Poissonian systems but, surprisingly, also increase single cell variability when the system is sub-Poissonian. While many studies developed analytical tools to study noise in single cells, there is a substantial gap in understanding the impact of cell-cell interactions on noise propagation. This well-written paper bridges this gap and will be extremely relevant to researchers interested in tissue modelling. I have some comments that could improve the paper:

1) In the results, when describing Figure 1 the authors should back up their statements that "there is no significant difference between the population/tissue level trajectories" with numbers (provide the simulation noise levels in the two cases).

2) The authors should add the derivation of the Fano Factors of the two systems studied (system (1) and (2)) in their Methods. Also to make the paper more accessible they should add the derivations of χ^2 and η^2 for these systems (page 4 right column).

3) Throughout the paper the authors analyze homogeneous systems; however, the biological relevance of their approach could be further strengthened by modelling more complex non-homogenous tissues. One way to do this is by studying the effect of transport on the noise of two types of cells with differing single-cell noise levels. For example, consider a field of cells implementing the 2-state model in Fig. 4 (Equation 7), composed of two types of cells that differ in their burst frequencies and consequently in their single-cell noise levels. The first type is a noisy cell type, e.g. with $K_{on}, K_{off} < d_0$ and the second is a non-noisy cell type, e.g. $K_{on}, K_{off} > d_0$ (for fair comparison one should ensure similar steady states level, e.g. $K_{on}/(K_{on}+K_{off})$ remains constant). How would the noise levels of each group be affected by the tissue coupling? A sparse noisy sub-population could represent a scenario of tissue ageing (e.g. PMID 16791200). A continuous patch of noisy cells could represent an aberrant proliferating tumor clone. There could be other ways to simulate such inhomogeneity, I think that this topic is worth including as it could demonstrate how tissue interactions could potentially keep renegade cells in check.

4) PMID 26828110 has single cell expression trajectories in an intact mammalian tissue, this could be an additional example worth analyzing to demonstrate the fit of their modeling approach to the experimental data.

Minor comments:

5) The statement that tissue-level data is frequently too low-resolution to distinguish individual cell outlines is not true, e.g. PMID 25728770.

6) A particular form of "transport" in tissues is cellular polyploidy, which has been shown to reduce noise in super-Poissonian systems, it is worth discussing this (PMID 25728770, PMID 23953111).

7) "Orange" in the figures looks like red.

Response to Referees

We would like to thank both referees for their detailed comments which have substantially improved the quality of the manuscript.

As a result of the reviews, we have changed the manuscript substantially. Principally, we have reframed our results in terms of single-cell variability rather than single-cell noise. There are a few reasons for this change. First, we felt - especially in light of the reviewer comments - that concepts such as “single-cell noise” and “tissue-level noise” are quite obscure and unintuitive, and particularly the word “noise” has more than one common definition. Second, we felt that the term “single-cell variability” better reflects the fact that our results have implications for the heterogeneity of tissues (a question brought up by reviewer 2, albeit in a slightly different context). Mathematically, we have defined a non-dimensional statistical measure of single-cell variability (L), that we believe is easier to understand than our previous based on our results regarding the unintuitive measures η^2 and χ^2 . Our results remain essentially the same, though we now have a greater focus on how cell-cell coupling affects heterogeneity.

Additionally, we have added two new graphs (Fig 5 (a) and (b)) in which we apply our technique to a dataset (suggested by reviewer 2) which uses paracrine signalling and is not oscillatory (as requested by reviewer 1). The dataset contained data from one tissue with low cell-cell communication, and another tissue with moderate cell-cell communication. This proved to be an excellent way to test our method, and we find that it works exactly as expected.

We have also replaced the conformation change example with a protein dimerization example (3). The reason for this is that the previous manuscript contained no examples with bimolecular reactions, an issue which this example rectifies.

Other significant changes, in response to the referee comments, include new paragraphs discussing the waiting-time distributions of transport between cells, the effects of environmental noise, the extension of the theory to heterogeneous populations of cells, and the new data in Fig. 5. On top of this, several sections have been substantially rewritten to make their meaning clearer.

Detailed comments to the reviewers are given below, and changes are shown in red in the revised manuscript. We hope the reviewers now find our revised manuscript suitable for publication.

Reviewer #1

Is it a fair assumption that a complex process like transportation is exponentially distributed? Are the results dependent on this assumption?

This is a very interesting point. In our simulations we have assumed that transport between neighbouring cells can be modelled as a Poisson-process, i.e. with exponential waiting times. The reasoning behind this is that a large proportion of cell-cell transport occurs by pure diffusion (e.g. molecules as large as proteins diffuse between neighbouring plant cells via plasmodesmata), and exponential waiting times are the standard for spatially-discretised diffusion modelling (e.g. the Reaction Diffusion Master Equation) because this model converges to Brownian motion in the limit of small discretisations. Active transport is likely to be more complicated, and no single model will be able to capture every feature of the process, however previous models of active transport have used exponential waiting times for ease-of-modelling purposes. However note that our theoretical results do not depend on

the exponential waiting time assumption. This is because we derive expressions in the special cases where transport is slow (i.e. no transport), and when it is fast (i.e. infinitely fast transport). In both cases there is no waiting-time distribution, so the exponential assumption does not affect our results. We have added a paragraph to the “Illustrative examples” section discussing this issue.

Will it be more realistic to assume transportation between neighboring cells, rather than any two cells? Given the spatial structure of the tissue, a specific cell may only be able to communicate with a few cells in its vicinity. Unless this is taken into account, it appears that their formulation may not be able to quantify the noise in individual cells.

The reviewer is correct that cell-cell communication principally occurs between neighbouring cells. Indeed all of our stochastic simulations use neighbour-neighbour interactions. For example in the model for two-stage gene expression in a tissue shown in (2): $P_i \leftrightarrow P_j$ denotes the reversible movement of protein molecules from cell i to a neighbouring cell j . This type of transport is used for all simulated models shown in Figs 2, 3 and 4. These results agree very well with the slow and fast limits of our theory. We have added some sentences to the Theory section to make this clearer.

An exact similar results on increase and decrease in noise depending on Fano factors has also been shown in other context, see Burger et al, PRE 2012. Influence of decoys on the noise and dynamics of gene expression

We disagree that this article is related to our work. Burger et al consider a model of transcription factor binding/unbinding to DNA, and observe that the Fano factor (Variance/Mean) of the unbound transcription factor decreases as the number of binding sites increases (the authors also refer to the Fano factor as “noise”, while our definition of noise is Variance/Mean², which confuses matters). The paper by Burger et al does not refer to multicellular organisms, cell-cell transport, diffusion, space or noise (Variance/Mean²), so it is hard to see how it relates to our work. However, we recognise that these competing definitions of noise are likely to be confusing for readers, and so we have reformulated our results in terms of single-cell variability/heterogeneity rather than single-cell noise. We hope this better reflects the novelty of our results.

Although authors comment on noise in slow/fast transport, they do not comment upon the behavior at intermediate transport rates.

We do comment on intermediate transport rates, but initially our focus is indeed on the special cases of fast/slow transport. The reason for this fast/slow focus is that completely general results can be derived in these special cases. We subsequently use these results to infer what happens at intermediate transport rates, e.g. if the slow limit is below the fast limit, then we expect the noise to increase so it is somewhere between these limits for intermediate transport rates. This is confirmed by the experimental results towards the end of our paper. Also note that Figs 2, 3 and 4 show the variation of single-cell noise as a function of transport rates thus interpolating between the slow and fast transport rate limits. We have added some sentences to our “Theory” section to make this clearer.

Moreover, it will be useful to comment how different noise mechanisms (external disturbances vs. low copy effects) may change the results.

The reviewer raises an important point. Intrinsic noise (including low copy effects) is the type of noise considered in the manuscript. Noise due to environmental fluctuations was not considered. The reason is that environmental noise affects every cell in a population equally.

This means that if environmental fluctuations induce a change in the single cell concentrations, there will be a corresponding change in the whole-tissue concentrations, so that our equations retain their validity and our results are unchanged. We have added a paragraph to the “Discussion” section regarding this point.

I was surprised to see no mention of paracrine signaling, which is the canonical cell-to-cell communication. In this case secreted signals, activate expression in other cells that lead to more secretion, creating spatial feedbacks. It will be useful to comment on such cases, which are far more common than those mentioned

The reviewer is absolutely correct to note that paracrine signalling is more common than the experimental examples studied in the original manuscript. We have added references to paracrine signalling in our abstract and introduction, and we have added two examples of experimental data with paracrine signalling, and find that our theory fits well. These are in the new Fig 5(a) and (b).

The data analysis was confusing. My understanding is that the simple expression models don't show oscillations, and system being analyzed does oscillate. It is not clear from the analysis how the noise results are true for any architecture with any form of dynamics.

It is true that none of the theoretical examples showed oscillations (this is principally to keep the mathematics simple), but our results are general and apply equally to oscillating systems and non-oscillating systems. Our theory makes no assumptions about the underlying network – it just assume that the same network is present in each cell – hence the strength of the approach. We have, however, reorganised the “Experimental results” section to add two examples of non-oscillating experimental datasets (Fig 5), and some sentences to the end of the “Theory” section stressing that our work applies to oscillating networks.

Reviewer #2

In the results, when describing Figure 1 the authors should back up their statements that “there is no significant difference between the population/tissue level trajectories” with numbers (provide the simulation noise levels in the two cases).

Due to the reorganisation of the paper, the sentence and plots to which the reviewer refers are no longer present. This is because we have reformulated our results in terms of single-cell variability/heterogeneity, rather than the less intuitive “single-cell noise” and “tissue-level noise”.

The authors should add the derivation of the Fano Factors of the two systems studied (system (1) and (2)) in their Methods. Also to make the paper more accessible they should add the derivations of χ^2 and η^2 for these systems (page 4 right column).

We have since replaced the old system (2) with a new system (3). We have added the exact expressions for the mean and variance of system (1), and the FF of system (3) to the section “Verification of theory using stochastic simulations” (system (3) is too complicated to give mean and variance as well, but we have given a reference to a paper where they can be found). We have also cited a reference on moment closure, for readers who want to go into detail of the calculation of the variances.

Throughout the paper the authors analyze homogeneous systems; however, the biological relevance of their approach could be further strengthened by modelling more complex non-

homogenous tissues. One way to do this is by studying the effect of transport on the noise of two types of cells with differing single-cell noise levels. For example, consider a field of cells implementing the 2-state model in Fig. 4 (Equation 7), composed of two types of cells that differ in their burst frequencies and consequently in their single-cell noise levels. The first type is a noisy cell type, e.g. with $K_{on}, K_{off} < d_0$ and the second is a non-noisy cell type, e.g. $K_{on}, K_{off} > d_0$ (for fair comparison one should ensure similar steady states level, e.g. $K_{on}/(K_{on}+K_{off})$ remains constant). How would the noise levels of each group be affected by the tissue coupling? A sparse noisy sub-population could represent a scenario of tissue ageing (e.g. PMID 16791200). A continuous patch of noisy cells could represent an aberrant proliferating tumor clone. There could be other ways to simulate such inhomogeneity, I think that this topic is worth including as it could demonstrate how tissue interactions could potentially keep renegade cells in check.

This is a very interesting suggestion. We have studied the example suggested by the reviewer in detail and found some fascinating results. Extending our results generally to heterogeneous systems is certainly plausible, and we have made considerable inroads thanks to the reviewer's suggestion, however the mathematics required seems sufficiently complicated to warrant its own paper. The current manuscript is already close to the 5000 word limit for Nature Communications, and we do not believe that we could do justice to these complex systems in the remaining space. We have, however, added a paragraph to the discussion outlining how the results could be extended, and noting that this will be the subject of a future paper. We would like to especially thank the reviewer for this suggestion, however, and we hope it will make for a great paper.

PMID 26828110 has single cell expression trajectories in an intact mammalian tissue, this could be an additional example worth analyzing to demonstrate the fit of their modeling approach to the experimental data.

We would like to thank the reviewer for suggesting this dataset, since it contains data from a tissue with few gap junctions (E18.5) and data from a tissue with many gap junctions (P1.5), and therefore is a perfect test for our theory. If our theory is correct, we would expect the E18.5 data to be noticeably closer to the slow limit than the P1.5 data. We have now analysed both datasets and found that this is indeed the case. We have two new graphs in Fig 5 for these datasets. The datasets are also particularly nice as they are non-oscillatory, unlike the other datasets we studied.

The statement that tissue-level data is frequently too low-resolution to distinguish individual cell outlines is not true, e.g. PMID 25728770.

We have corrected this.

A particular form of "transport" in tissues is cellular polyploidy, which has been shown to reduce noise in super-Poissonian systems, it is worth discussing this (PMID 25728770, PMID 23953111).

We have mentioned this in the introduction.

"Orange" in the figures looks like red.

We have now referred to it as red everywhere.

REVIEWERS' COMMENTS:

Reviewer #2 (Remarks to the Author):

The authors have addressed most of my points and have significantly improved the paper. Also their format change makes the text much clearer. I recommend publication in Nature Communications.

Reviewer #3 (Remarks to the Author):

The authors have addressed the comments of reviewer 1 well. I really enjoyed reading the paper. I only have one very minor point. The legend in Figure 6 says single-cell data. This is incorrect, as this data is single-pixel.